# Pathogenic and Endosymbiotic Bacteria and Their Associated Antibiotic Resistance Biomarkers in *Amblyomma* and *Hyalomma* Ticks Infesting Nguni Cattle (*Bos* spp.)

**DOI:** 10.3390/pathogens11040432

**Published:** 2022-04-02

**Authors:** Aubrey Dickson Chigwada, Ntanganedzeni Olivia Mapholi, Henry Joseph Oduor Ogola, Sikhumbuzo Mbizeni, Tracy Madimabi Masebe

**Affiliations:** 1Department of Life and Consumer Sciences, College of Agriculture and Environmental Sciences, University of South Africa (UNISA), Florida Campus, Roodepoort 1709, South Africa; 61366943@mylife.unisa.ac.za (A.D.C.); maphon@unisa.ac.za (N.O.M.); henryogola@gmail.com (H.J.O.O.); mbizes@unisa.ac.za (S.M.); 2School of Agricultural and Food Sciences, Jaramogi Oginga Odinga University of Science and Technology, Bondo P.O. Box 210-40601, Kenya

**Keywords:** *Amblyomma*, *Hyalomma*, tick microbiome, 16S rRNA, antibiotic resistance markers, Nguni cattle, PICRUSt

## Abstract

Deciphering the interactions between ticks and their microbiome is key to revealing new insights on tick biology and pathogen transmission. However, knowledge on tick-borne microbiome diversity and their contribution to drug resistance is scarce in sub–Saharan Africa (SSA), despite endemism of ticks. In this study, high-throughput 16S rRNA amplicon sequencing and PICRUSt predictive function profiling were used to characterize the bacterial community structure and associated antibiotic resistance markers in *Amblyomma variegatum*, *A. hebraeum**,* and *Hyalomma truncatum* ticks infesting Nguni cattle (*Bos* spp.). Twenty-one (seven families and fourteen genera) potentially pathogenic and endosymbiotic bacterial taxa were differentially enriched in two tick genera. In *H. truncatum* ticks, a higher abundance of *Corynebacterium* (35.6%), *Porphyromonas* (14.4%), *Anaerococcus* (11.1%)*, Trueperella* (3.7%), and *Helcococcus* (4.7%) was detected. However, *Rickettsia* (38.6%), *Escherichia* (7%), and *Coxiellaceae* (2%) were the major differentially abundant taxa in *A*. *variegatum* and *A. hebraeum*. Further, an abundance of 50 distinct antibiotic resistance biomarkers relating to multidrug resistance (MDR) efflux pumps, drug detoxification enzymes, ribosomal protection proteins, and secretion systems, were inferred in the microbiome. This study provides theoretical insights on the microbiome and associated antibiotic resistance markers, important for the design of effective therapeutic and control decisions for tick-borne diseases in the SSA region.

## 1. Introduction

Ticks are important arthropods that act as vectors of various bacterial communities infecting cattle. The enormous annual global loss of about US$22 to US$30 billion has been recorded in livestock production due to tick-borne pathogens; therefore, ticks and tick-borne disease control are very important in animal health and meat production [1,2,3]. Several studies in South Africa have identified ticks of the genera *Ixodes, Hyalomma*, *Amblyomma*, and *Rhipicephalus* infesting Nguni cattle [4]. *Hyalomma* and *Amblyomma* tick-associated pathogens include bacterial species in the genus *Anaplasma*, *Borrelia*, *Coxiella*, *Ehrlichia*, *Francisella*, and *Rickettsia* [5,6,7,8,9,10]. The *Coxiella burnetii*, *Ehrlichia ruminantium*, and *Rickettsia rickettsia* were found most prevalent in *Amblyomma* ticks, collected from cattle in Cape Town, South Africa [5]. Similarly, ticks collected from dogs in the North West, KwaZulu-Natal, Mpumalanga, and Free State South African provinces were dominated by pathogenic species of *Coxiella*, *Anaplasma*, *Rickettsia* bacterial genera [8].

Diseases associated with tick-borne bacteria include relapsing fever (*Borrelia burgdorferi, B. afzelii,* and *B. garinii*) and spotted fever (*Rickettsia rickettsia*); ovine and bovine anaplasmosis (*Anaplasma ovis* and *A. marginale*); Q fever (*Coxiella burnetti*); ehrlichiosis; and heartwater (*Ehrlichia ruminantium*) [11,12,13,14]. Antibiotics such as tetracyclines, macrolides, beta-lactams, aminoglycosides, and fluoroquinolones are used for the treatment of infections caused by tick-borne bacteria. On the other hand, bacterial antibiotic resistance can be achieved through mechanisms employing over-expression of efflux pumps, under-expression of porins, iron transport proteins, and enzymes that modify or degrade antibiotics, and chromosomal mutations and mutations in drug target sites have been reported [15,16]. The genes encoding for methylases (*erm*), drug efflux pumps [*mef*(A)], and ribosomal mutations in 23S rRNA have been shown to confer macrolide resistance; these have been identified in tick-borne pathogens such as *Ehrlichia chaffeensis*, *Ehrlichia canis* [17], *Anaplasma phagocytophilum* [18], *Franscisella tularensis* [19], *Rickettsia typhi,* and *Rickettsia prowazekii* [20]. Several studies have shown tetracycline resistance in tick-borne bacteria *F. tularensis* [21,22], and this resistance is attributed to genes coding for active overexpressed efflux pumps [*tet*(A-H)], ribosomal protection subunits [*tet*(M-P)], and drug modifying enzymes [*tet*(X)]. Furthermore, resistance to beta-lactams was reported in *Francisella* species attributed to genes that coded for *bla*A, *amp*G protein, and metallo-β-lactamase [23], while class C β-lactamase enzymes have also been detected in *Rickettsia felis* and *R. conorii* strains [24,25]. Despite increasing concerns of drug resistance development in tick-borne pathogens, there is limited information on the tick microbiome ecology and their involvement in drug resistance in local African cattle breeds.

In addition to the pathogens they transmit, several studies have reported that ticks can also harbor several symbiotic and commensal microbes that may be key to vector competency and pathogen transmission dynamics [26,27]. These include Coxiella-like endosymbionts (CLEs), Franscisella-like endosymbionts (FLEs), Rickettsia-endosymbionts, and Wolbachia-like and other commensal tick microbes that are members of the phylum Proteobacteria, Firmicutes, Bacteroidetes, and Actinobacteria [5,27]. Both the pathogenic and endosymbiotic bacteria coexist within the tick, with increasing evidence that interaction between tick-borne pathogens and tick microbiome is bidirectional [28,29,30]. To the tick host, nonpathogenic endosymbionts and commensals may confer multiple detrimental, neutral, or beneficial effects related to fitness, nutritional adaptation, development, reproduction, defense against environmental stress, and immunity [26,27]. On the other hand, non-pathogenic microorganisms may also play a role in driving the transmission of tick-borne pathogens (TBP) [31]. Despite the accumulating evidence on the link between tick microbiome, tick biology, and tick-borne disease dynamics, the microbiome of many tick species, particularly those that are not common human disease vectors, have yet to be investigated widely in Africa. Few available studies have reported microbial communities in whole intact ticks without consideration of organ-specific community distributions [32]. Salivary glands and mouthparts of ticks serve as routes for efficient pathogen transmission and maintenance of endosymbionts [33,34]. Organ-specific studies are also important in characterizing microbes transmitted, acquired, and maintained within the salivary glands and mouthparts [34]. Studies focusing on antibiotic resistance biomarkers are also limited. We anticipate that comparisons of microbiome compositions and endosymbiont patterns between tick species may offer valuable information for better understanding how tick microbiomes are shaped, how they influence vector competency, and tick-borne pathogenesis [30]. 

Currently, metagenomics using high-throughput Illumina technology and pyrosequencing enables the routine, comprehensive characterization of microbial communities from diverse environments using culture-independent methods. For example, analysis of 16S rRNA gene amplicon sequences has become the standard method for culture-independent studies of tick microbial diversity [32,35]. Furthermore, several 16S rRNA gene studies have extended the ability to infer the functional contribution of individual bacterial community members by mapping a subset of abundant 16S rRNA sequences to their nearest sequenced reference genomes. Towards this, predicting microbial functions from 16S rRNA gene sequencing data is currently a common alternative to shotgun metagenomic approaches. Phylogenetic Investigation of Communities by Reconstruction of Unobserved States (PICRUSt) constitutes a novel computational algorithm that enables the prediction and establishment of protein and metabolic function profiles based on the frequency of detected 16S rRNA sequences of bacteria corresponding to genomes in regularly updated, functionally annotated genome databases [36]. The ability of PICRUSt to infer metabolic information in genomes included in databases such as the Kyoto Encyclopedia of Genes and Genomes (KEGG) based on reference phylogenetic trees of 16S rRNA gene amplicons has made it a popular prediction tool for metagenomic function. Several studies have proven the effectiveness of the PICRUSt algorithm in the characterization of functional and resistance biomarkers of intracellular bacteria under different environments, which is usually impossible when using cultural techniques [37,38]. Bioinformatic approaches including predictive metagenomic profiling using PICRUSt have also been applied in the study of tick microbiomes [29,31,39]. As inferences based on predicted functional traits may suffer from inherent inaccuracies in resolving functional biogeography in certain ecosystems [36], validation of the PICRUSt annotation is always very important. Recently, two studies validated by PCR the functional predictions of PICRUSt annotation on tick microbiomes [40,41], providing support for the use of PICRUSt2 as a suitable tool to accurately predict functional perturbations in the tick microbiome.

In Southern Africa, the indigenous Nguni cattle, an admixture of hump-less zebu (*Bos taurus*) and humped zebu (*Bos indicus*), is one of the largest breeds owing to their adaptation to suboptimal environmental conditions, including less susceptibility to tick infestation [42,43]. Despite the higher levels of resistance to tick infestation and tick-borne diseases reported in Nguni cattle than other breeds, the tick challenge is still major [44]. In this study, we explored the composition and structure of the bacterial community associated with *Amblyomma (**A. variegatum* and *A. hebraeum)* and *Hyalomma truncatum*) tick species infesting Nguni cattle in South Africa using high-throughput 16S rDNA amplicon sequencing on an Illumina MiSeq platform. To gain insight on functional profiles and antibiotic resistance, the prediction tool PICRUSt was used to determine the functional resistance biomarkers of the bacterial communities in the two tick species. We envisaged that comprehensive characterization of both culturable and unculturable bacterial communities, including their antibiotic resistance and disease pathogenesis biomarkers, may provide a deeper understanding of the tick-borne pathogens. This information can be key towards the better elucidation of possible recommendations for strengthening programs to prevent and control the potential infections caused by tick-borne pathogens in the region.

## 2. Results

### 2.1. Global Sequencing Data and Tick Microbiome Diversity

A total of 1,065,139 quality sequence reads comprising 1214 operational taxonomic units (OTUs) were generated. Among the results, sample (H2) had 0.0003% valid reads and was thus excluded in the downstream analysis. Overall, the average Good’s coverage of the library ranged between 99.34% to 99.90%. In addition, rarefaction curves plots approached plateaus or asymptotes with increasing sample size (Figure A1), suggesting sequencing depth was adequate to reliably describe the bacterial microbiome associated with the tick genera. In this study, exploratory analyses based on alpha and beta diversity, including multivariate analyses did not observe any significant difference between *A. variegatum* and *A. hebraeum* community diversity and composition structure (Figure A2). Therefore, for downstream analyses, *A. variegatum* and *A. hebraeum* samples were collectively grouped as *Amblyomma* ticks and compared with *H. truncatum* samples, herein referred to as *Hyalomma* ticks.

Comparative analysis of the alpha diversity and species richness indices between *Hyalomma* and *Amblyomma* bacterial communities is illustrated in Figure 1. Wilcoxon rank-sum test revealed no significant differences in observed OTUs, Chao1, and ACE indices after multiple testing corrections (Kruskal-Wallis, FDR > 0.05), but alpha diversity indices Shannon, Simpson, Inv Simpson, and Fisher diversity showed a significant difference between *Hyalomma* and *Amblyomma* tick microbiome (pairwise Wilcoxon rank-sum test, *p* = 0.021). This suggested significant differences in species richness and genetic composition of bacterial communities associated with *Hyalomma* and *Amblyomma* tick species.

For a glimpse of compositional and structural similarity of tick bacterial communities in samples, we employed beta diversity analysis based on Jaccard indices. The principal-coordinate analysis (PCoA) revealed a significant difference in beta-diversity (*p* < 0.05), with samples showing separation and clustering of samples into two groups according to tick species (Figure 2). The total x-axis variances PCoA 1 was 38.6% and the y-axis PCoA 2 was 13.1%, with prediction ellipses observed having tick species falling in different ellipses, reflecting subtle variances in the associated bacterial communities is dependent on tick species. Further, both analysis of similarity (ANOSIM) and permutational multivariate analysis of variance (PERMANOVA) showed that bacterial composition patterns differed significantly according to tick species (adonis PERMANOVA, *F* = 5.21, *p* = 0.021; ANOSIM, *R* = 0.321, *p* = 0.011).

### 2.2. Microbial Community Structure

At the phylum level, classified sequence reads revealed four major phyla: Actinobacteria (34.65%), Proteobacteria (31.41%), Firmicutes (23.40%), and Bacteroidetes (9.37%), with the remaining phyla accounting for <1.15% of total abundance (Figure 3a). However, members of phylum Proteobacteria were relatively abundant in *Amblyomma*, while Actinobacteria, Bacteroidetes, and Firmicutes were dominant taxa in *Hyalomma* tick species. To further delineate the differences in bacterial community composition and structure, a two-sided *t*-test statistical analysis coupled with multiple test correction Storey FDR (false discovery rate) at a 95% confidence interval was performed. An extended error plot (Figure 3b) revealed significant differences (*q*-value < 0.0001) in mean proportions in *Amblyomma* and *Hyalomma* bacterial communities at the phylum level. Members of phylum Proteobacteria were significantly enriched in *Amblyomma* ticks, whereas Firmicutes, Bacteroidetes, and Actinobacteria were the significant taxa in *Hyalomma* ticks.

At the genus level, *Amblyomma* tick species had a comparatively high abundance of *Rickettsia* (38.6%), *Escherichia* (7%), *Arthrobacter* (3.6%), and *Coxiella* (2%), while *Hyalomma* tick species had a high abundance of *Corynebacterium* (35.9%), *Porphyromonas* (14.4%), *Anaerococcus* (11.1%), *Trueperella* (3.7%), and *Helcococcus* (4.7%) (Appendix A). Further, a dendrogram heatmap plot (Figure 4) showing the relatedness of bacterial communities at the genera level revealed clustering together of *Amblyomma* tick samples in the ordination space. In contrast, *Hyalomma* tick species clustered with two *Amblyomma* samples A2 and A11, suggesting shared genera or co-occurrence of some bacterial communities in both tick species. Overall, the top genera identified in this study were *Rickettsia*, *Corynebacterium*, *Porphyromonas*, *Trueperella*, Coxiellaceae_uc, etc, (Figure 4, Appendix A).

In this study, species-level microbiome analysis using EzBioCloud 16S Database (www.ezbiocloud.net, accessed on 28 February 2022) [45] to identify the tick microbiome was also undertaken (Figure A3). The most dominant species in *Amblyomma* ticks were *Rickettsia rickettsia*, *Escherichia coli*, *Aerococcus vaginalis*, Coxiella_uc group, and *Acinetobacter globiformis*. In contrast, *Corynebacterium* group (*C. xerosis*, *C. falsenii*, *C. resistens*, *C. striatum*, *C. epidermidicanis*, and *C. pseudotuberculosis*), *Porphyromonas levii*, *Trueperella pyogenes*, JQ480818_s (*Coxiella* endosymbiont) were identified in *H. truncatum***.**

### 2.3. Core Microbiome and Metagenomic Biomarker Identification

A Venn diagram was defined as core OTUs at genus level present in at least 50% of the samples of each group at 1% minimum relative abundance and used to evaluate the similarities between *Amblyomma* and *Hyalomma* bacteria; the diagram is illustrated in Figure 5a. Overall, 74.4% OTUs of the core microbiome was shared between tick species, while 6.3% and 8.9% OTUs were found to be unique to *Amblyomma* and *Hyalomma* ticks, respectively. About 10.6% of non-core microbial bacteria were identified. A summary of the top 30 shared and unique genera representing the core microbiome in the two tick species is presented in Appendix A. To further investigate the taxonomic apportionment and detect differentially abundant taxa, we compared the abundance of the unique core OTUs at the family and genus levels. The resultant taxonomic profile was then used by LEfSe to detect metagenomic biomarkers. Overall, LEfSe detected 21 differentially abundant biomarkers(LDA > 2.0, *q* < 0.01) including seven family and fourteen genera level biomarkers across the two tick species (Figure 5b,c). The largest number of taxonomic biomarkers was detected in *H. truncatum* ticks, with genera ascribed to phylum Actinobacteria (*Corynebacterium*, *Trueperella*, and *Tessarococcus*), Bacteriodetes (*Porphyromonas*), Firmicutes (*Anaerococcus*, *Helcococcus*, *Peptoniphilus*, *Peptococcus*, *Finegoldia*, and unclassified Peptoniphilacae), and Proteobacteria (*Coxiella*) as the key taxa. In contrast, only proteobacterial genera *Rickettsia* and *Escherichia* were identified as biomarkers in *A. variegatum* and *A. hebraeum* ticks.

### 2.4. Distribution of Potentially Pathogenic Taxa in the Tick Samples 

As ticks are known to be the most important vectors of pathogens [1,2], this study also examined the distribution of potentially pathogenic bacterial taxa associated with two tick species. Potential Pathogenic genera in the top 40 abundant OTUs identified included *Rickettsia*, *Ehrlichia*, *Coxiella*, *Porphyromonas*, *Trueperella*, *Corynebacterium*, and *Helcococcus*. Interestingly, these taxa, with exception of genus *Ehrlichia* and *Bacillus*, constituted the metagenomic biomarkers detected by LEfSe analysis (Figure 5b,c). We then performed the two-sided White’s non-parametric *t*-test to identify differences in the pathogenic microbiome between *Amblyomma* and *Hyalomma* ticks. Consistent with LEfSe results, potentially pathogenic genus *Rickettsia* was exclusive and highly (*q*-value < 0.0001) abundant (accounting for 36.7% of sequence reads) in *Amblyomma* tick samples (Figure 6). Other differentially abundant (White’s non-parametric test, *p* < 0.05) potentially pathogenic genera in both *A. variegatum* and *A. hebraeum* included unclassified Coxiellaceae and *Escherichia*. In contrast, *Porphyromonas*, *Trueperella*, *Corynebacterium*, *Coxiella*, and *Helcococcus* were highly enriched (*q*-value < 0.0001) in *H. truncatum* ticks.

### 2.5. Prediction of the Functional Profiles in the Tick Microbiome

In order to gain insight on the metabolic contribution to antibiotic resistance and disease pathogenesis, the prediction tool PICRUSt2 was used to determine to reveal the functional differences in terms of metabolic, antibiotic resistance, and disease pathogenesis (virulence) biomarkers of the bacterial communities between the two tick species. A total of 28 KEGG pathways showing distinct abundance between *Amblyomma* and *Hyalomma* ticks are illustrated in an extended error plot in Figure 7a. The principal pathways such as metabolism, genetic information processing, environmental information processing, metabolism, and cellular information processing pathways, including human disease pathways, were common to both *Amblyomma* and *Hyalomma* microbiomes. Despite subtle differences in the abundance of functional pathways, no significant differences (R^2^ = 0.957) were detected in the overall composition of the two tick species (Figure 7b).

One of the key abundant pathways included the genetic information processing related to the biogenesis of ribosomal protection proteins, protein sorting, protein export, and aminoacyl-tRNA biosynthesis. Similarly, metabolic pathways relating to amino acid metabolism, degradation, and biosynthesis of enzymes and secondary metabolites such as streptomycin, penicillin, and cephalosporin were also highly expressed in both tick species. These pathways may account for potential enzyme-derived antibiotic resistance and drug degradation in bacteria communities detected in ticks. Environmental information processing pathways such as membrane transport proteins and efflux pumps, secretion systems, and phosphotransferase enzymes (two-component systems, phosphatidylinositol, MAPK signaling, and bacterial toxins) and the cellular processing pathways (porins regulation and inorganic ion transport) that may be key in bacterial pathogenesis [46] were also enriched.

### 2.6. Drug Resistance and Disease Pathogenesis Biomarker Analysis

For understanding the drug resistance and disease pathogenesis potential of the tick-borne bacterial community, the predicted functional profiles were subjected to LEfSe analysis to detect differentially expressed drug resistance and pathogenesis biomarkers. In all, LEfSe detected 116 KEGG orthologs (KOs) as differentially (LDA score > 2, *p* < 0.05) enriched functional biomarkers (Appendix A). Comparing *Amblyomma* and *Hyalomma* tick bacterial communities, LEfSe identified 50 significant drug resistance and pathogenesis biomarkers that were differentially enriched between the tick species (Figure 8). 

Overall, *Amblyomma* and *Hyalomma* tick bacterial communities had 34 and 16 antibiotic resistance and pathogenesis (virulence) markers, respectively, that were differentially enriched (Figure 8a). Supporting LEfSe results, a scatter plot showed poor correlation (*R^2^* = 0.0001, *p* < 0.05), indicating significant differences in the compositional diversity of drug resistance and pathogenesis biomarkers in *Amblyomma* and *Hyalomma* microbiomes (Figure 8b). The main classes of KO genes associated with drug resistance that were differentially expressed included genes coding for drug efflux pumps, drug degrading and modifying enzymes, secretion system proteins, and ribosomal protection proteins. Specifically, MFS efflux pumps such as MHS family transporter genes encoding alpha-ketoglutarate permeases (K033761) and proline/betaine transporters (K03762), the PAT family gene encoding for beta-lactamase induction signal transducer *Amp*G (K08218), and the DHA2 family gene encoding for multidrug resistance proteins (K03446) were inferred in both tick microbiomes. Whereas K033761 (*p* = 0.036), K03762 (*p* = 0.012), and K08218 (*p* = 0.008) were significantly enriched in *H. truncatum*, the DHA2 (K03446) and MATE family multidrug resistance proteins (K03327) including multidrug efflux pumps (K18138, K18139, K03543, and K07799) were differentially (*p* < 0.05) abundant in both *A. variegatum* and *A. hebraeum* microbiomes. Further, drug antiporters in the *NhaA* family such as Na+:H+ antiporters (K03313) and metal resistance genes involved in ATP-binding protein systems for the iron complex and peptide/nickel transport (K02032, K02031, and K02035) and Cu^+^-exporting ATPase (K17686) were also highly enriched in both *A. variegatum* and *A. hebraeum* microbiomes.

Most importantly, inferred functional KO of bacterial communities from *Hyalomma* and *Amblyomma* microbiomes also revealed the presence of drug resistance enzymes. Specifically, penicillin degrading enzymes such as the beta-lactamases class C (K01467) and D (K17838), as well as the penicillin inhibiting and modifying enzymes guanylyltransferase (GTase) (K00971) and 3-demethylubiquinone-9 3-methyltransferase (K00568) were identified. Furthermore, enzymes conferring ribosomal resistance to macrolides such as 23S rRNA (adenine2030-N6)-methyltransferase (K07115) and 3-deoxy-D-manno-octulosonic-acid transferase (K02527), including ribosomal protection protein biosynthesis enzymes such as GTP diphosphokinase and guanosine-3′,5′-bis (diphosphate) 3′-diphosphatase (K01139) were also detected. These enzymes may play an important in ribosome-linked drug resistance in the tick microbiome. A prominent observation was that four of these resistance enzymes (beta-lactamase classes C (K01467) and D (K17838), 3-demethylubiquinone-9,3-methyltransferase (K00568), 23S rRNA (adenine2030-N6)-methyltransferase (K07115), and 3-deoxy-D-manno-octulosonic-acid transferase (K02527)) were more highly enriched in the *Amblyomma* than the *Hyalomma* microbiome.

In addition to drug resistance enzymes, drug detoxification enzymes such as the glutathione S-transferases (K00799), known to inhibit the MAP kinase pathway, and the antitoxin YefM proteins (K19159) involved in modulation of toxins as well as environmental stress, were also significantly enriched (*p* < 0.05) in both *Amblyomma* microbiomes. Finally, genes coding for enzymes and proteins involved in virulence and pathogenesis such as *AraC* family transcriptional regulator (K03755) and bacterial transpeptidases (Sortase A) (K07284), as well as versatile type IV secretion system proteins *vir*B4, *vir*B6, *vir*B8, *vir*B9, and *vir*B11 (K03196, K03199, K03201, K03203, and K03204, respectively) were also inferred (Figure 8a, Appendix A). These are protein complexes normally powered by ATP to secrete protein toxins key to pathogenesis, bacterial survival as well as drug resistance.

## 3. Discussion

In the current study, host ticks from the genera *Amblyomma* (*A. variegatum* and *A. hebraeum*) and *Hyalomma* (*H. truncatum*) infesting indigenous Nguni cattle in the Roodaplate ARC research farm were collected between September 2018 and February 2019. Previous studies have confirmed the presence of these tick species, including *Rhipicephalus* (*Boophilus*) ticks, in South African cattle [4,6,8,47,48]. However, there is a paucity of information on the microbiome associated with host ticks infesting Nguni cattle. There is accumulating evidence that vector-borne infections in the vertebrate host are shaped by the microbiome of the arthropod vector and its competence to acquire and maintain infections with vector-borne pathogens [26,27,30]. In this study, we described taxonomic and functional characteristics of the microbiome associated with *Amblyomma* and *Hyalomma* ticks infesting indigenous Nguni cattle and infer several potential taxonomic, drug resistance, and pathogenesis (virulence) markers that may help in deciphering tick-borne disease dynamics in Nguni livestock.

Ticks are important ectoparasites that are characterized by a complex and dynamic microbial community, ranging from vertically-transmitted pathogenic symbionts to transient commensals acquired from the local environment, that are key to their host interactions, survival, and disease transmission [26]. Hard ticks such as *Amblyomma* and *Hyalomma* tick species are known to harbor *Coxiella*-like endosymbionts (CLEs), *Franscisella*-like endosymbionts (FLEs), *Rickettsia*-endosymbionts, and *Wolbachia*-like and commensal tick microbes that are members of the phylum Proteobacteria, Firmicutes, Bacteroidetes, and Actinobacteria [5,49]. Consistent with these findings, the most dominant phyla identified in this study were Proteobacteria, Firmicutes, Bacteroidetes, and Actinobacteria with a total of 612 bacterial genera identified in the two tick species. The dominance of bacterial taxa belonging to the genus *Rickettsia*, *Corynebacterium*, *Porphyromonas*, *Trueperella*, *Helcococcus*, and *Actinomycetospora* was observed. Overall, we found that community alpha diversity did not vary among the tick species; however, the species richness (Figure 1) and beta diversity (Figure 2) were lower in *Amblyomma*. Interestingly, the aggregated mean Shannon diversity index was low (ranging between 1.76 to 5.72) (Figure 1), suggesting shared bacteria genera among the tick species. This is consistent with findings reported elsewhere [32,39,50], where a few core bacteria taxa, likely endosymbionts, dominate the tick microbiome. Our results also showed that *A. variegatum* and *A. hebraeum* samples grouped apart from *H. truncatum,* revealing distinct microbial community structure. This was further supported by multivariate analyses that revealed that bacterial composition patterns differed significantly according to tick species (adonis PERMANOVA, *F* = 5.21, *p* = 0.021; ANOSIM, *R* = 0.321, *p* = 0.011). These observations are in conformity with previous microbiome ecological studies in tick species [5,6,10] reporting that tick identity, the presence of cattle host blood engorgements, feeding habits, shape and size of mouthparts, and geographical location of the tick samples, as well as a previous tick host, greatly influence microbial community structure [30]. 

An analysis of community composition showed that *A. variegatum* and *A. hebraeum* ticks presented a significantly higher abundance of *Proteobacteria* mainly ascribed to the genus *Rickettsia* when compared to *H. truncatum* samples (Figure 3b). *Rickettsia* was detected in all the *Amblyomma* samples and accounted for 36.7% of all sequence reads that were identified to belong to *Rickettsia rickettsia,* providing proof that *A. hebraeum* and *A. variegatum* ticks may be the principal vector of *Rickettsia*-like symbionts in Nguni cattle. Interestingly, *Rickettsia* and *Escherichia* were identified as the key metagenomic biomarkers at the genus level (Figure 5b), indicating their importance in *Amblyomma* tick interactions with host Nguni cattle, their survival, and disease transmission. Magaia et al. [51] reported that 80% of *A. hebraeum* ticks in cattle in Mozambique were infected by *R*. *Africae,* while Jongejan et al. [52] reported a higher abundance of *R. Africae* in adult and nymph *A. hebraeum* ticks in goats in Mpumalanga Province, South Africa. In that study, the high relative abundance of *Rickettsia* species in nymphs also provide clues for their vertical transmission from egg masses. These observations imply that, in addition to tick species, host species and geographical location also play a significant role in modulating the tick microbiome. 

Generally, Rickettsia endosymbionts are obligate intracellular gram-negative bacteria that play a major role in tick physiology and survival; for example, several Rickettsia phylotypes have abilities to synthesize folate, which supplements tick nutrition due to lack of this essential vitamin in the blood meal [30,53]. In addition, *Rickettsia* is associated with zoonotic diseases such as spotted fever and typhus groups [6,53]. Specifically, the findings that high abundance of *R. rickettsia* group across all *Amblyomma* tick samples, even from the relatively tick-resistant Nguni cattle breed, confirm their wider presence in South Africa, as has been previously reported in other cattle breeds and tick species [8,9,43,44]. This information is particularly important to tourists and travelers visiting South Africa as it is related to the risk of tick bites and the potential of rocky mountain spotted fever infections. 

In this study, we also anticipated a higher percentage of family Anaplasmataceae in the *Amblyomma* ticks, based on previous studies that have reported co-infection of *Ehrlichia ruminantium* and *R. africae* in *A. hebraeum* ticks [52], and a higher abundance of genus *Anaplasma* in ticks infesting Nguni cattle [3,43]. The only other Rickettsia-like endosymbionts member of family Anaplasmataceae detected included genus *Ehrlichia*; however, it was only observed in 22% of *Amblyomma* samples and constituted 0.2% of total sequence reads. Lack of detection could be attributed to the differences in the methodology used and target tick species, where higher abundance in *Rhipicephalus* ticks than other species has previously been reported in South Africa [44]. Pathogenic *Ehrlichia ruminantium*, which are agents of heartwater (ehrlichiosis), were detected in *Amblyomma* and *Hyalomma* tick species. However, the low abundance compared to other pathogenic groups may account for the lower seroprevalence and incidence of ehrlichiosis and heartwater that have been previously reported in Nguni cattle [44].

In *Hyalomma* ticks, members of phyla Firmicutes (*Anaerococcus*, *Helcococcus*, *Peptoniphilus*, *Peptococcus*, *Finegoldia*, and unclassified Peptoniphilacae), Actinobacteria (*Corynebacterium*, *Trueperella*, and *Tessarococcus*), Bacteriodetes (*Porphyromonas*), and Proteobacteria (*Coxiella*) were highly enriched, with these taxa also identified as the key metagenomic biomarkers (Figure 5b), as well as tick-borne pathogens specific to *Hyalomma* ticks (Figure 6). Similar to Rickettsia endosymbionts, Coxiella-like symbionts are obligate intracellular gram-negative bacteria vertically transmitted in ticks and are primary endosymbionts with a major role in B vitamin supplementation, nutrients missing from the host’s blood [26]. Their involvement in reproductive fitness in *Amblyomma americanum* has also been reported [54]. Consistent with our findings, several studies have recorded the presence of pathogenic *Coxiella burnetii* in all tick species identified in South Africa [5,8,14]. These findings are also in agreement with other reports where high serological indices of *Coxiella* were detected all over Africa, mainly in tick species of *Amblyomma*, *Hyalomma*, and *Rhipicephalus* [51,55]. In this study, sequence reads of *Coxiella* identified were mostly of uncultured bacterial strains; therefore, future studies for further isolation, sequencing, and in-depth characterization of these bacteria are warranted.

Other potentially pathogenic commensals identified in *Hyalomma* ticks were taxa ascribed to genus *Corynebacterium*, *Helcococcus*, *Arthrobacter*, *Porphyromonas*, *Anaerococcus*, *Aerococcus*, *Peptoniphilus*, *Tessarococcus*, and *Trueperella*. Interestingly, all these taxa were also detected in *Amblyomma* samples, albeit at lower abundance, indicating that they are environmentally acquired. These groups have been reported to be associated with ruminant blood, ticks, and other sources [53,56,57]. *Corynebacterium* are normal flora of animal skin with some species identified as opportunistic pathogens causing zoonotic diseases. Zoonotic species include *C. pseudotuberculosis* associated with secondary meningitis, caseous lymphadenitis, and otitis media-interna in cattle and goats [57]. *C. xerosis* has also been linked to abscesses in the brain, mastitis, osteomyelitis, abortions, and arthritis, while *C. falsenii*, *C. bovis*, *C. resistans* and *C. striatum* causes the mouth of an eagle, mastitis, bronchial aspirates, and blood culture and abscess, respectively [56,57,58]. In this study, the presence of *C. xerosis*, *C. falsenii*, *C. resistens*, *C. striatum*, *C. epidermidicanis*, and *C. pseudotuberculosis* was detected in both tick species with a higher abundance of these species observed in *Hyalomma*. It is plausible that these species are innocuous microbiomes of animal skin that are acquired by ticks during feeding. On the other hand, *Trueperella pyogenes* is an opportunistic pyogenic infectious agent mainly found in the mucus that causes otitis externa, abortions, metritis, infertility, and mastitis in cattle [59,60]. Although data on ticks as vectors of *T. pyogenes* are scarce, Rzewuska et al. [60] reported tick’s contribution in the transmission of *T. pyogenes*. Finally, members of the genus *Porphyromonas* are emerging animal and human pathogens with species such as *P. levii* implicated in bovine necrotic vulvovaginitis in cattle [61]. In this study, *P. levii* species were identified in both *Hyalomma* and *Amblyomma*, implicating these tick species as reservoirs and potential vectors for the emerging pathogen. The identification of commensals alongside the known Coxiella-like endosymbionts (*Coxiella*) as key metagenomic biomarkers gives clues on their importance in H. truncatum biology.

Coupled with the microbiota characterization, we used 16S rDNA sequencing data to predict metagenome functions and used the inferences of microbial function to another dimension in characterizing the differences of the microbiota between *Amblyomma* and *Hyalomma*. The accuracy of these predictions is measured by the nearest sequenced taxon index (NSTI), which estimates how closely related the microorganism in the studied samples are to microorganisms with already sequenced genomes. In this study, the NSTI values for *Amblyomma* and *Hyalomma* samples of 0.14 ± 0.08 and 0.19 ± 0.03, respectively, were comparable to values reported for soil (NSTI = 0.17) and human microbiome samples (NSTI = 0.03) [62]. Overall, the compositional diversity of KEGG level 2 pathways related to metabolism, information processing, environmental information processing, and cellular information processing was similar in both *Amblyomma* and *Hyalomma* microbiomes (Figure 7), despite subtle differences observed in the relative abundances.

In this study, we focused on several gene families (KOs) of medical importance relating to antimicrobial resistance and diseases pathogenesis markers such as drug efflux pumps, drug degrading and modifying enzymes, secretion system proteins, and ribosomal protection proteins that were differentially enriched in the two tick microbiomes (Figure 8). These observations support previous in silico findings analyzing resistance genes in tick-borne bacteria [25,63,64]. The most abundant efflux pumps inferred included the Major Facilitator Superfamily (MFS) transporters, ATP-Binding Cassette type 2 (ABC-2), and the Multidrug and Toxic Compounds Extrusion (MATE) family multidrug resistance (MDR) proteins (Figure 8a), whose role in bacterial antibiotic resistance has been widely described [63,65,66]. Comparatively, the *Amblyomma* microbiome exhibited a significantly higher abundance of ABC-type 2 transporters and MATE family MDR efflux pumps, and a concomitant higher abundance of genus *Rickettsia* than *Hyalomma* microbiome (Figure 6). Consistent with our findings, Rolain [67] reported that *Rickettsia* resistance to macrolides and beta-lactam antibiotics could be linked to the overexpression of ABC-type 2 multiple drug transport systems. Curiously, only MFS efflux pumps ascribed to the Metabolite:H+ Symporter (MHS) family were highly enriched in the *Hyalomma* microbiome (Figure 8a). Such proteins are integral membrane transporters linked to tetracycline and quinolones resistance mechanisms in *Coxiella burnetti* [68]. Furthermore, MFS multiple drug antiporters of the *Nha*A family such as Na+:H+ antiporters were also present in both microbiomes. These genes have previously been identified in the *Coxiella* genus, and have been implicated to confer resistance to fluoroquinolones [69]. The differential overexpression of the MDR efflux pumps genes inferred in this study provide clues on their importance in drug resistance (tetracycline, quinolone, and fluoroquinolone). Further, they represent novel targets for drug development; specifically, designing a new class of peptidomimetic efflux pump competitive inhibitors for improvement of veterinary and medical treatment for tick-borne bacterial infections is an attractive strategy. However, successful deployment of such treatment will require further pharmacodynamics and pharmacokinetics studies to determine the efficacy and safety of combined administration of efflux pump inhibitors and antibiotics in the management of tick-borne bacterial pathogens.

An array of metal resistance genes such as ATP-driven iron complex transport and peptide/nickel transport, and Cu^+^-exporting ATPase, a selection factor that may be critical for the proliferation of co-resistance mechanisms for heavy metals and antibiotics in bacterial pathogens [70], was also identified. These proteins were highly enriched in the *Amblyomma* microbiome, indicating potential cross-resistance mechanisms for metal and antibiotic resistance in this tick microbiome. Moreover, versatile type IV secretion systems (*vir*B4, *vir*B6, *vir*B8, *vir*B9, and *vir*B11) involved in protein toxin production, including *Ara*C family transcriptional regulator and bacterial transpeptidases that are key to bacterial virulence and pathogenesis [71,72], were enriched in *Amblyomma* and *Hyalomma* microbiomes. Another key observation was the abundance of genes encoding drug degrading and modifying enzymes such as the beta-lactamases classes C and D, as well as the penicillin inhibiting and modifying enzymes guanylyltransferase (GTase) and 3-demethylubiquinone-9 3-methyltransferase linked to penicillin resistance in tick-borne *Ehrlichia* [73], *Rickettsia* [74], and *Corynebacterium* [56] in both tick microbiomes. Finally, ribosome and protein synthesis represent one of the major targets in the bacterial cell for clinically-relevant antibiotics such as macrolides (e.g., erythromycin). The macrolide target ribosomal-site-altering enzymes such as 23S rRNA (adenine2030-N6)-methyltransferase (*RlmJ*) and 3-deoxy-D-manno-octulosonic-acid transferase (*kdtA*), have been reported to confer macrolide resistance [75] and virulence [76]. Macrolide resistance is attributed to the alteration or mutation of 23S ribosomal RNA and methylation of the domain V of 23S rRNA by methyltransferase enzymes [75]. Additionally, macrolide ribosomal protection proteins and biosynthesis enzymes such as GTP diphosphokinase or guanosine-3′,5′-bis (diphosphate) 3′-diphosphatase were also detected. The current findings of the abundance of these proteins may be linked to the richness of genus *Rickettsia* and *Ehrlichia* in *Amblyomma* microbiomes, which is consistent with the detection of macrolide resistance in *Rickettsia*, *E. chaffeensis*, *E. canis*, *Anaplasma phagocytophilum*, and *Francisella tularensis* tick-borne pathogens [17,25].

In this study, all antibiotic resistance biomarkers were mainly ascribed to potentially pathogenic genera in the three tick microbiomes. However, there is accumulating evidence that all pathogenic, commensal, as well as environmental bacteria form a reservoir of antibiotic resistance genes (the resistome) from which pathogenic bacteria can acquire resistance via horizontal gene transfer [77]. Thus, it is plausible that the indiscriminate use and misuse of antibiotics for the management of tick infestation and tick-borne diseases make animal hosts and tick vectors a potential hotspot for the dissemination of antibiotic resistance in the tick microbiome. Under such scenarios, resistance genes, mobile genetic elements (MGEs), and (sub-inhibitory) antibiotic selection pressure from various sources may be introduced to endosymbionts, commensals, and pathogens [78]. Hence, further in-depth studies are needed to help understand the extent of the resistomes and how their mobilization in pathogenic bacteria in cattle breeds may occur under tick infestation endemisms [78].

In conclusion, this study identified the key differences in endosymbiotic community diversity and inferred antibiotic resistance and pathogenesis in *Amblyomma* (*A. variegatum* and *A. hebraeum*) and *Hyalomma* (*H. truncatum*) microbiomes in Nguni cattle. However, whether the endosymbionts and commensals detected in this study are active and their exact involvement in tick physiology and pathogen acquisition, nutrition, and environmental adaptability need to be addressed in the future in order to construct a more holistic view of the tick microbiome. Another limitation of the current study was that all the inferences were based on predicted functional traits by PICRUSt annotation, which may suffer from inherent inaccuracy in resolving functional profiles in certain ecosystems [62]. Therefore, further in-depth validation through functional assays using a large sample size will be important. Nevertheless, our data contribute to the growing knowledge on the link between the key metagenomic/taxonomic taxa associated with *A. variegatum, A. hebraeum,* and *H. truncatum* microbiomes and inferred antibiotic resistance and pathogenesis biomarkers that may provide valuable theoretical insights on tick biology, the ever-changing epidemiology of tick-borne diseases and future drug discovery.

## 4. Materials and Methods

### 4.1. Study Site, Tick Collection, and Identification

Ticks used in this study were collected between September 2018 and February 2019 from the Roodeplaat ARC-research farm, Gauteng, South Africa (29°59″ S, 28°35″ E). To collect the ticks, tweezers were used to remove ticks from cattle, ensuring the mouthparts remained intact. Ticks were then placed into Eppendorf test tubes containing 70% ethanol for preservation. The cattle bite site was carefully cleaned with 70% ethanol. Collected ticks were then transported immediately to the University of South Africa Eureka Life Science Laboratory and stored at −80 °C for subsequent identification and DNA extraction. From the 100 cattle sampled, a total of 110 ticks were collected and identified morphologically to species level using standard taxonomic identification keys as previously described [79,80]. A total of 19 ticks (15 and 4 *Amblyomma* and *Hyalomma* samples, respectively) were used for further analysis.

### 4.2. Sample Preparation for Microbiome Analysis 

After microscopic identification and confirmation, ticks were washed with nuclease-free water to remove ethanol, then air-dried. Ticks were then cut under a light microscope from the second leg up to the capitulum to target the salivary glands [10]. The upper sections having salivary glands were cut into pieces and added to 0.5 mL screw-cap tubes. The omega TL^®^ lysis buffer and 25 μL of Proteinase K were added to each tube and lysed for a 24-h incubation at 56 °C. DNA extraction was performed using the E.Z.N.A.^®^ tissue DNA extraction kit, (Omega Bio-Tek, Inc., Norcross, GA, USA), according to the manufacturer’s instructions.

### 4.3. Library Preparation and 16S rRNA Metagenomic Sequencing

Library preparation for the 16S rRNA targeted amplicon sequencing was performed according to the protocol described by Ogola et al. [81]. Briefly, the DNA samples were amplified targeting the V3-V4 hypervariable region of the 16S rRNA using universal primers 27 F (5′-TCGTCGGCAGCGTCAGATGTGTATAAGAGACAGCCTACGGGNGGCWGCAG-3′) and 518 R (5′-GTCTCGTGGGCTCGGAGATGTGTATAAGAGACAGGACTACHVGGGTATCTAATCC-3′) primers with overhang adapters (underlined). The PCR reaction mixture (25 μL) comprised of 2.5 μL of DNA, 12.5 μL of 2x KAPA HiFi Hot Start Ready Mix (Kapa Biosystems, Wilmington, MA, USA), and 5 μL of each of the primers. The thermocycling conditions used included: initial denaturation at 94 °C for 5 min, 36 cycles of denaturation at 94 °C for the 30 s, annealing at 58 °C for 30 s, and elongation at 72 °C for 40 s and final elongation at 72 °C for 10 min before infinite cooling at 4 °C. The resulting amplified products were visualized in ethidium bromide-stained 1% agarose gel. The DNA pools that yielded amplified products with fragments of approximately 560 bp were selected. The DNA was quantified with a Qubit^®^ fluorometer (Life Technologies Carlsbad, Carlsbad, CA, USA) using a Qubit dsDNA HS^®^ Assay kit (Thermofisher Scientific Corporation, Waltham, MA, USA). Subsequently, the amplified products were cleaned using Ampure XP beads (Beckman Coulter, Brea, MA, USA), 80% EtOH, and magnetic beads following manufacturer’s instructions. The resultant purified products were attached to dual indices using the Nextera XT v2 Index Kit [39]. Briefly, a total reaction mixture of 25 μL comprising 5 μL DNA, 2.5 μL each of Nextera index primers, 12.5 μL of 2x KAPA HiFi Hot Start Ready Mix (Kapa Biosystems, Boston, MA, USA), and 2.5 μL of PCR grade water was prepared. The reaction mixture was amplified under the following thermocycling conditions: initial denaturation at 95 °C for 3 min, 8 cycles of denaturation at 95 °C for 30 s, annealing at 55 °C for 30 s, elongation at 72 °C for 40 s and final elongation at 72 °C for 10 min before holding infinitely at 4 °C. The amplified products were purified using Ampure XP beads, 80% EtOH, and magnetic beads following manufacturer’s instructions. Quantification of the final product was performed using Qubit. The concentrated final library samples were diluted to 4 nM using 10 mM Tris at pH 8.5. A volume of 5 μL of each sample was pooled into a multiplexed library and a negative control sample was included. The 6 pM of the pooled libraries and the PhiX control library were denatured using diluted 0.2 N NaOH to achieve cluster generation during sequencing according to the manufacturer’s protocol (Illumina Inc., San Diego, CA, USA).

The final library was sequenced by paired-end (300 bp reads) sequencing v.3 chemistry along with its multiplex sample identifiers on the Illumina MiSeq Platform (Illumina Inc., San Diego, CA, USA) according to standard protocol. The dataset for this study was submitted to NCBI under Bio-project PRJNA753497 with accession numbers SAMN20695925 to SAMN20695943.

### 4.4. Bioinformatic Processing of 16S rRNA Amplicon Sequencing Data

Sequence processing was performed using Mothur (version 14) software as per Miseq SOP [82]. The SILVA-based reference sequences (Silva v132) were used to classify unique sequences by executing a Bayesian classifier on the Mothur platform. UCHIME algorithm was used with the Silva SEED database to identify and remove chimeras for downstream analysis, while a rarefaction was used to remove singletons [83]. Using the average-neighbor algorithm, the classified 16S rRNA was assigned operational taxonomic units (OTUs) at 97.0%. Generated OTU tables were then used for downstream analysis, where R (version 4.0.3) and STAMP (version 2.1.3) software were used for statistical data analysis and visualization as previously described [84].

Briefly, OTU tables were rarefied and normalized to the lowest number of read count of 31,484 reads. In this study, we observed no significant differences in the alpha diversity and bacterial composition structure of *A. variegatum* and *A. hebraeum*. Therefore, all samples of *A. variegatum* and *A. hebraeum* were collectively grouped as *Amblyomma* samples, while all *H. truncatum* samples were here referred to as *Hyalomma*. Alpha diversity was evaluated based on observed OTUs, Chao1, ACE indices, Shannon, Simpson, Inv Simpson, and Fisher diversity; significant differences were calculated using a Storey false detection rate (FDR)-corrected pairwise Wilcoxon rank-sum test. Beta diversity was analyzed by microbial principal coordinate analysis (PCoA) based on bacterial Jaccard distances. To complement beta diversity analysis, we also used anosim and adonis functions in the vegan package to perform analysis of similarity (ANOSIM) and permutational multivariate analysis of variance (PERMANOVA) with 999 permutations based on Bray-Curtis distances to evaluate the contribution of the tick species to the bacterial composition and structure. These analyses were carried out with the “vegan” package in R. Significances for differences in the abundances of taxa were determined based on the storey FDR-corrected two-sided White’s nonparametric *t*-test with 1000 permutations implemented in STAMP software. The heatmap of the relative abundances of the top 50 genera and Venn diagram of the core microbiome at the genus level (OTUs present in at least 50% of the samples of each group at 1% minimum relative abundance) were generated using *heatmap.2* and *ampVis2* packages in R. LEfSe was also used to elucidate the biomarkers in each group.

### 4.5. Metagenomic Prediction of Functional Resistance biomarkers

Metabolic and resistance biomarkers were predicted by PICRUSt v2 algorithm software using 16S rRNA sequence data and reference databases to infer biomarker gene contents as described by Douglas et al. [36]. Using the PICRUSt v2 algorithm, COG (Cluster of Orthologous Genes), and KEGG databases, resistance biomarkers were identified to Level 2 Orthology. We also calculated the weighted Nearest Sequenced Taxon Index (NSTI), a measure of the availability of nearby genome representatives for the given OTUs, to assess the overall feasibility of the PICRUSt approach. The differential abundance of predicted KO genes was evaluated by LEfSe analysis [85] under the EzBiocloud MTP pipeline [45]. For this analysis, the alpha parameter significance threshold for the Kruskal–Wallis (KW) test implemented among classes in LEfSe was set to 0.01 and the logarithmic LDA score cut-off was set to 2.0.

## Figures and Tables

**Figure 1 pathogens-11-00432-f001:**
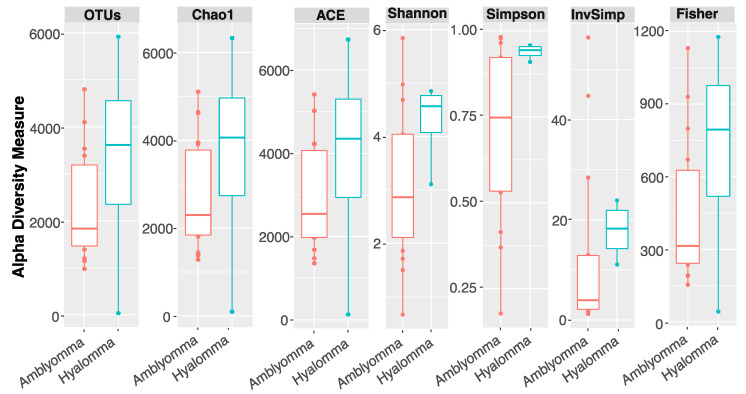
Alpha diversity analysis. Boxplots show Shannon, Simpson, Inv Simpson, Fisher diversity, observed OTUs, Chao1, and ACE indices. The diversity indices Shannon, Simpson, Inv Simpson, Fisher results showed significant differences between *Amblyomma* (*A. variegatum* and *A. hebraeum*) and *Hyalomma* (*H. truncatum*) (pairwise Wilcoxon rank-sum test, *p =* 0.021), but the richness indices, observed OTUs, Chao1, and ACE showed no significant differences after multiple testing corrections (Kruskal–Wallis, FDR > 0.05).

**Figure 2 pathogens-11-00432-f002:**
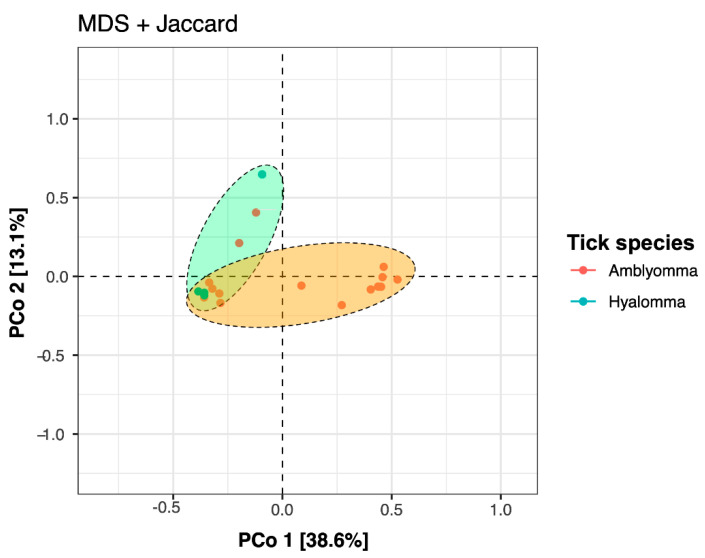
Principal coordinate analysis (PCoA) plot showing relatedness between *Amblyomma* (*A. variegatum* and *A. hebraeum*) and *Hyalomma* (*H. truncatum*) tick bacterial community structures at the genus level. The PCoA was based on multidimensional scaling (MDS) and Jaccard distances and the ellipses represent the 95% confidence based on a multivariate t-distribution.

**Figure 3 pathogens-11-00432-f003:**
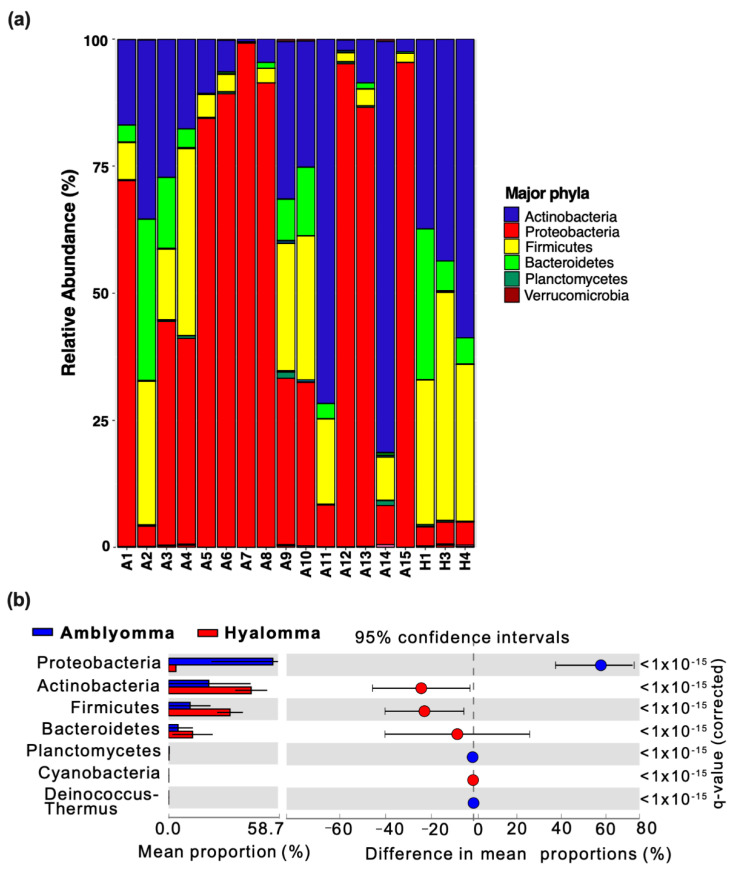
Relative abundance of bacterial communities at the phylum level. (**a**) Stacked bar chart representing the taxonomic bacterial composition at the phylum level. Samples A1-15 and H1-3 denote *Amblyomma* (*A. variegatum* and *A. hebraeum*) and *Hyalomma* (*H. truncatum*) ticks, respectively. (**b**) Bacterial phyla that were differentially enriched in the *Amblyomma* and *Hyalomma* tick species. Each extended error bar plot indicates the *p*-value along with the effect size and the associated difference in the mean proportion and confidence interval for each phylum. Each bar plot indicates the mean proportion of OTUs assigned to the phylum in each group. q-values represent *p*-values obtained by White’s nonparametric *t*-test and Storey FDR correction.

**Figure 4 pathogens-11-00432-f004:**
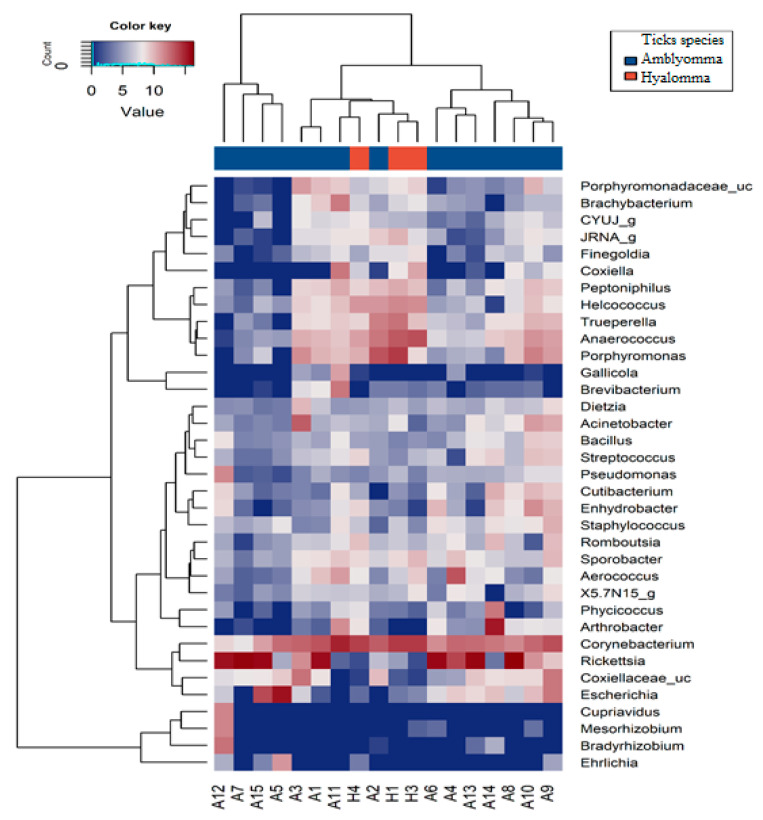
Heatmap of the 35 most abundant genera of bacteria communities in *Amblyomma* (*A. variegatum* and *A. hebraeum*) and *Hyalomma* (*H. truncatum*) tick species. The dendrogram shows complete-linkage agglomerative clustering based on Euclidean distance. The heatmap color (blue to reddish-brown) represents the row z-score of the mean relative abundance from low to high.

**Figure 5 pathogens-11-00432-f005:**
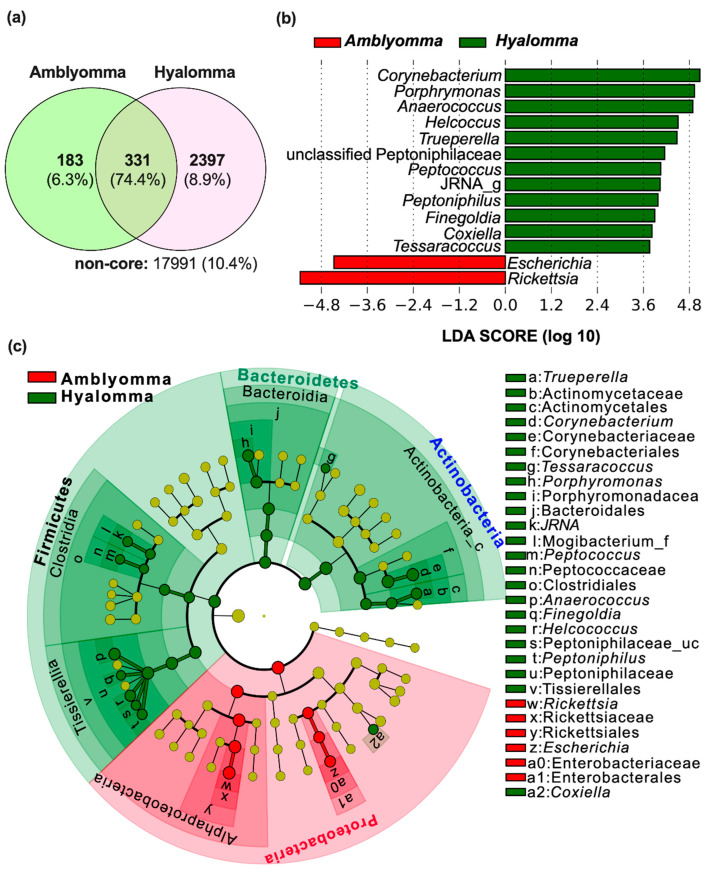
Core microbiome and metagenomic biomarkers in the tick microbiome. (**a**) Venn diagram showing the shared and unique core microbiome at the genus level (OTUs present in at least 50% of the samples of each group at 1% minimum relative abundance). (**b**) Linear discriminant analysis (LDA) effect size (LEfSe) plot depicting the differentially taxonomic/metagenomic biomarkers in *Amblyomma* (*A. variegatum* and *A. hebraeum*) and *Hyalomma* (*H. truncatum*) microbiome at a logarithmic LDA score > 2. (**c**) LEfSe generated cladogram of the taxonomic/metagenomic biomarker differences in the two tick microbiomes.

**Figure 6 pathogens-11-00432-f006:**
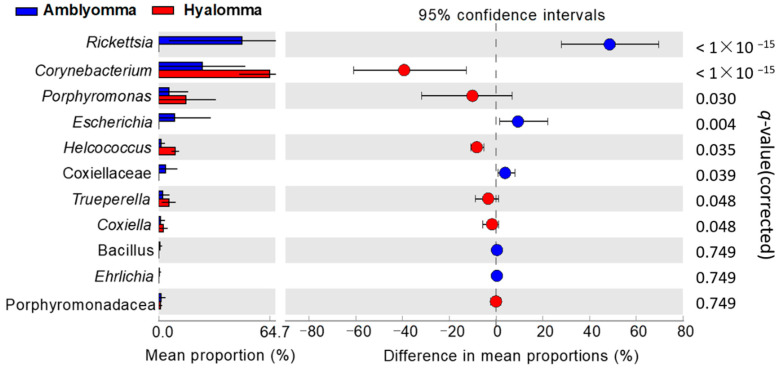
Potentially pathogenic taxa that were differentially enriched in the tick samples. Extended error plot illustrating eleven potentially pathogenic bacteria at the genus level that were differentially abundant between *Amblyomma (**A. variegatum and A. hebraeum)* and *Hyalomma (H. truncatum*) ticks, as tested by a two-sided White’s nonparametric *t*-test. FDR-adjusted *p* values are reported at the right of the image.

**Figure 7 pathogens-11-00432-f007:**
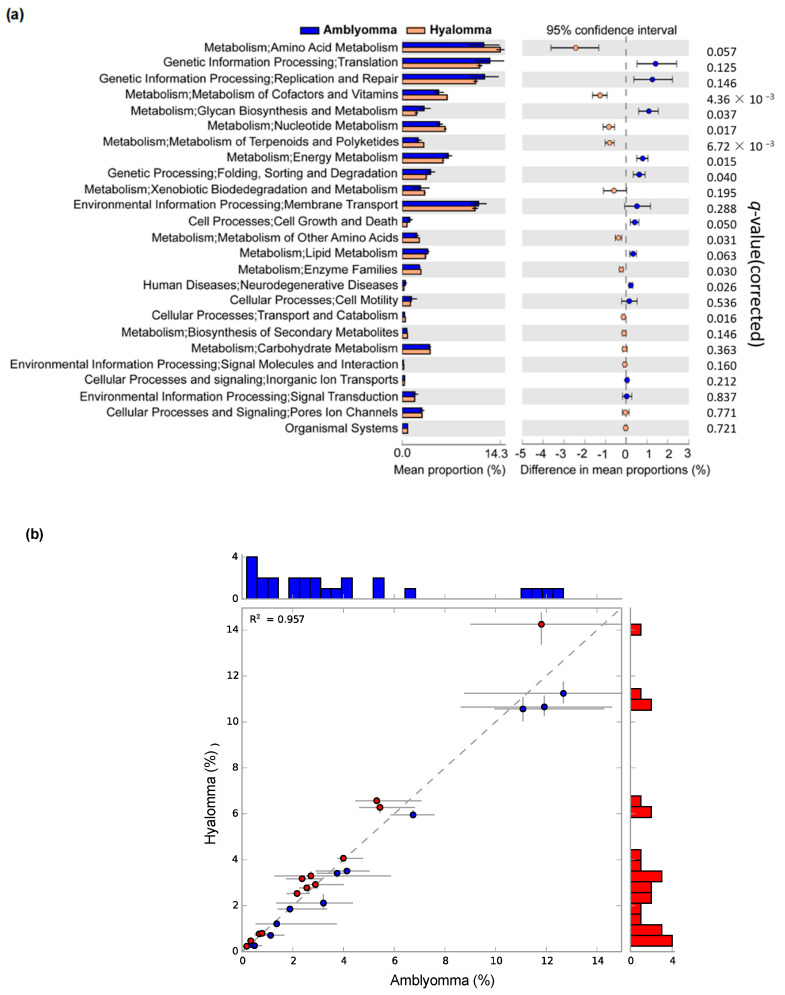
The major KEGG pathways in the tick microbiome. (**a**) Differential PICRUSt predicted KEGG pathways between tick microbiomes detected by STAMP software. The *p*-values (adjusted by Benjamini–Hochberg correction to account for false discovery rates), effect size, and 95% confidence interval bootstrap computed by White’s non-parametric *t*-test (two-sided type) are indicated. (**b**) Scatter plot showing the correlation of the predicted functional genes in the bacterial community in *Amblyomma (**A. variegatum* and *A. hebraeum)* and *Hyalomma (H. truncatum*) ticks. White’s non-parametric *t*-test using bootstrap dissimilarity showed that clusters were significant at (R^2^ = 0.957, *p* < 0.05).

**Figure 8 pathogens-11-00432-f008:**
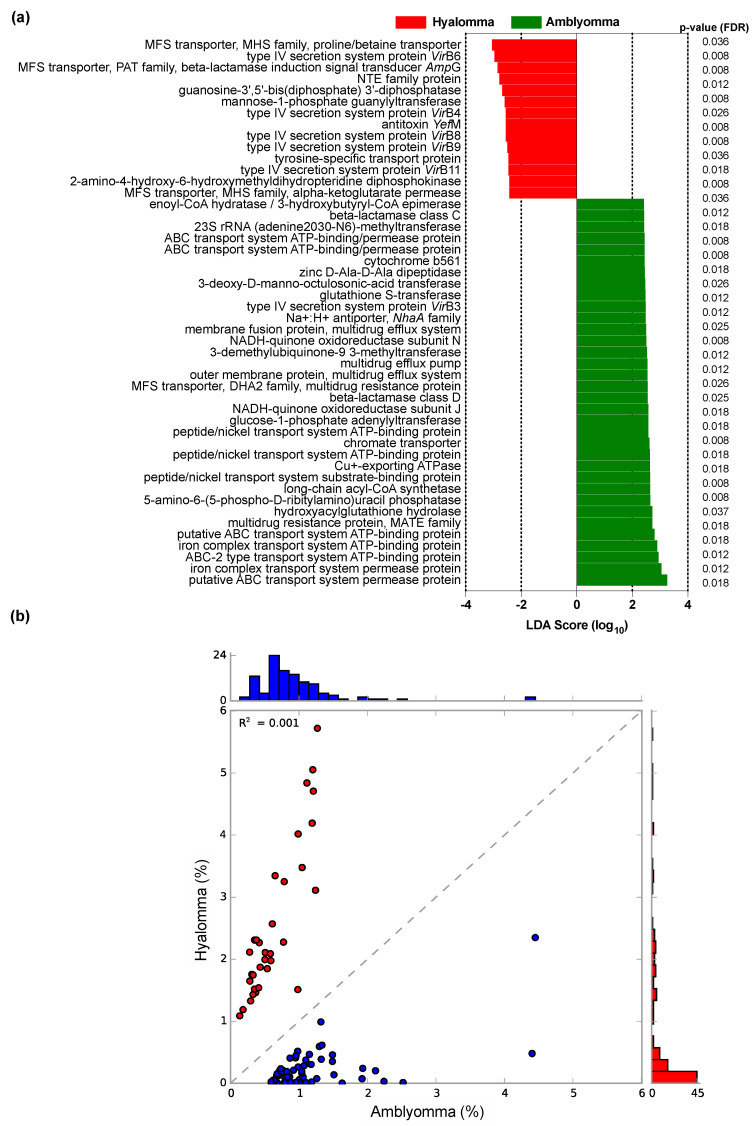
Differentially abundant KEGG orthologs related to drug resistance and pathogenesis biomarkers in *Amblyomma* (*A. variegatum* and *A. hebraeum*) and *Hyalomma* (*H. truncatum*) tick microbiomes. (**a**) LEfSe histogram of the differential drug resistance and pathogenesis biomarkers at a logarithmic LDA score > 2. The *p*-values (adjusted by Benjamini–Hochberg correction to account for false discovery rates) are shown. (**b**) Scatter plot showing clustering of the biomarkers of bacteria in the two tick species. The plot used a two-sided, White’s non-parametric *t*-test at 95% confidence interval with the DP bootstrap method.

## Data Availability

Not applicable.

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
