# Peer review of "Pathogenic and Endosymbiotic Bacteria and Their Associated Antibiotic Resistance Biomarkers in Amblyomma and Hyalomma Ticks Infesting Nguni Cattle (Bos spp.)"

_pathogens, 2022, doi:10.3390/pathogens11040432_

Round 1

Reviewer 1 Report

The manuscript describes the diversity of Amblyomma (A. variegatum and A. hebraeum) and Hyalomma (H. truncatum) ticks microbiome by 16S rRNA sequencing. The metagenomic function assay also provides clues for the main microbiome and their associated antibiotic resistance markers.

Overall, the manuscript is well written, the experimental design and the materials and methods applied very well support the results and conclusion of the study. I have only a suggestion for the authors should consider. I wonder if the title should be more specific to better reflect the results obtained in this work.

Author Response

Comments and Suggestions

The manuscript describes the diversity of Amblyomma (A. variegatum and A. hebraeum) and Hyalomma (H. truncatum) ticks microbiome by 16S rRNA sequencing. The metagenomic function assay also provides clues for the main microbiome and their associated antibiotic resistance markers. Overall, the manuscript is well written, the experimental design and the materials and methods applied very well support the results and conclusion of the study. I have only a suggestion for the authors should consider. I wonder if the title should be more specific to better reflect the results obtained in this work.

We are humbled and grateful for your efficient work the in-depth and insightful reviewing and processing of the draft that has greatly help to shape the article. We revised the manuscript title to be more specific as suggested by the reviewer  (Line 2-4)

Reviewer 2 Report

The study presents an innovative idea that has been poorly explored in previous studies (that of the presence of Antibiotic Resistance Biomarkers in the tick microbiome). Major revisions are needed to reach the level required for publication. Specific comments were provided below so the authors can improve their work.

Comments

General: Language and syntaxes of sentences require major revisions. Another important point is that, while the authors used two Amblyomma (A. variegatum and A. hebraeum) and one Hyalomma species (H. truncatum) in their study (lines 20-21 and 382-383), the results are not referred to tick species, but genera. This has to be addressed in a revised version on the manuscript (including all sections: Introduction, Results, Discussion and methods) because, as the authors probably know, the tick microbiome is different from species to species.

Specific comments

Title: The title is too long. Please trim it and focus on the most relevant message.

Abstract

Line 13: The expression ‘Deciphering ticks and their microbiomes’ needs revision. What does ‘Deciphering ticks and/or their microbiomes’ means?

Line 14: Change ‘tickborne’ to ‘tick-borne’ and revise this in the whole manuscript.

Lines 13-15: Sentence too long. Split it.

Line 16: ‘evolving drug resistance’ needs revision. It has no clear meaning.

Lines 18-19: Remove the full name of ‘PICRUSt’.

Line 20: ‘antibiotic resistance’ is trait intrinsic to some bacteria expressing ‘antibiotic resistance’ genes. In this study, the authors studied the presence of ‘antibiotic resistance’ functions in the microbiome of ticks. ‘antibiotic resistance’ lacks meaning here. Please, correct.

Lines 20-21: the fragment ‘antibiotic resistance in Amblyomma … and Hyalomma … microbiome in Nguni cattle’ needs revision. I suggest change to ‘antibiotic resistance in the microbiome of Amblyomma … and Hyalomma … ticks collected in Nguni cattle’.

Line 22: distinctions between ‘pathogenic and endosymbiotic bacteria’ cannot be made using the genera level as the taxonomic resolution. Please, revise. In almost any given bacterial genus (e.g., Rickettsia), you can find ‘pathogens, commensals and/or endosymbionts’. Almost none bacterial genus has only ‘pathogens’.

Lines 21-25: In this sentence is not clear what bacterial taxa belongs to ‘Hyalomma’ and which one to ‘Amblyomma’. Also, as two ‘Amblyomma’ species were studied, please, change to ‘Amblyomma spp.  microbiomes’

Line 25: This study was not designed to identify ‘taxonomic biomarkers’. Remove.

Line 29: What would the ‘key microbiome’ be? What the ‘non-key microbiome’ be?

Introduction

Line 39: Change ‘ticks of the genera Ixodes…’

Line 40: Doing what ‘in Nguni cattle’? infesting? Feeding on?

Line 41: Species the name of the pathogen species.

Line 42: The ‘R’ in Rickettsia goes in Italics too.

Lines 47-49: Mention the species of the pathogens you are referring to.

Line 51: Change ‘pathogens’ to ‘bacteria’.

Lines 51-55: This sentence is not the contraposition of the previous one, nor presents an exception from hat was stated in the previous one. Therefore, remove ‘However’. Instead, ‘On the other hand’ may be better.

Lines 57-59: Species should be named in full the first time they are mentioned in the text. Then, the abbreviated form can be used.

Lines 65-67: Change to ‘Despite increasing concerns of drug resistance development, there is limited information on the tick microbiome ecology and their involvement in drug resistance of tick-borne bacteria affecting local African cattle breeds.’

Lines 74-75: ‘barriers for … pathogen transmission’? Something that contributes to the transmission should not be regarded as a ‘barrier’.

Line 81: Chance to ‘competence’ and ‘tick-borne pathogen pathogenesis’

Lines 91-101: It could be mentioned that previous studies used PICRUSt to study the tick microbiota should be mentioned in the introduction. What type of studies have used this methodology applied to tick microbiota? (e.g., see Pathogens. 2020 ;9(4):309. doi: 10.3390/pathogens9040309 and Microorganisms. 2020;8(11):1829. doi: 10.3390/microorganisms8111829.). Validation of PICRUSt predictions is also important. At least two studies have validated by PCR the functional predictions of PICRUSt on tick microbiome (Front Immunol. 2021;12:704621. doi: 10.3389/fimmu.2021.704621 and Vaccines (Basel). 2020;8(4):702. doi: 10.3390/vaccines8040702.). This further justifies the use of PICRUSt.

Results

Line 165: ‘p’ in italics.

Line 167: Not clear why the authors performed this analysis at the ‘phylum’ level. Why not using a lower taxonomical category (e.g., genera)?

Lines 185-186: Percentage values can be organized from higher to lower or lower to higher. Also, taxonomic levels above ‘genera’ do not go in Italics (please revise this in the whole manuscript).

Lines 226-229: To what tick genera/species these results refer to.

Line 231: Here is important to specify the proportion of samples in which the ‘core’ taxa were found. It means that if, for example, only 1% of Amblyomma sp. samples have one given taxon, this cannot be considered as ‘core’ within that tick species. In other words, identification of a taxonomic core would require also presence across samples under study (see e.g., Pathogens. 2020 ;9(4):309. doi: 10.3390/pathogens9040309). In addition, defining a ‘core’ microbiome considering the small ‘n’ of the study is questionable.

Line 259: Here as in (line 22) please consider that distinctions between ‘pathogenic and endosymbiotic bacteria’ cannot be made using the genera level as the taxonomic resolution. Please, revise. In almost any given bacterial genus (e.g., Rickettsia), you can find ‘pathogens, commensals and/or endosymbionts’. Almost none bacterial genus has only ‘pathogens’.

Discussion

General

The discussion (specifically from line 511) misses a crucial point of addressing the possibility that antibiotic resistance enzymes found in tick microbiome may be coded by genes that could be transferred to tick-borne bacteria via horizontal gene transfer or any other molecular mechanism accounting for genetic material transference in Prokaryotes.

Material and methods

Figure

Figure 1. Change ‘Sample size’ to ‘Number of reads’.

Figure 3. The labels of the axis are not legible. Please, provide a corrected figure.

Figure 4a. Despite the legend shows several taxa/colors, only four (blue, yellow, red and green) are visible in the bars. Then it is not clear the proportion of the four in taxa in relation to the others.

Figure 6. Genera names go in Italics.

Figure legends

Provide a short title for each figure and then follow with the explanation.

Author Response

Comments and Suggestions

The manuscript describes the diversity of Amblyomma (A. variegatum and A. hebraeum) and Hyalomma (H. truncatum) ticks microbiome by 16S rRNA sequencing. The metagenomic function assay also provides clues for the main microbiome and their associated antibiotic resistance markers.

Overall, the manuscript is well written, the experimental design and the materials and methods applied very well support the results and conclusion of the study.

Authors Response: We are humbled and grateful for your efficient work in-depth and insightful reviewing and processing of the draft that has greatly helped to shape the article. We have thoroughly revised the manuscript based on each suggestion or comment of the reviewer. 

I have only a suggestion for the authors should consider. I wonder if the title should be more specific to better reflect the results obtained in this work.

We revised the manuscript title to be more specific as suggested by the reviewer  (Line 2-4)

Language and syntaxes of sentences require major revisions.

We apologize for our previous carelessness. We have corrected the typographical errors in the full text. In addition, the professional language editor has helped to check and modify the potential typographical errors, and improve the readability and nuances as suggested by the reviewer.

Another important point is that, while the authors used two Amblyomma (A. variegatum and A. hebraeum) and one Hyalomma species (H. truncatum) in their study (lines 20-21 and 382-383), the results are not referred to tick species, but genera. This has to be addressed in a revised version on the manuscript (including all sections: Introduction, Results, Discussion and methods) because, as the authors probably know, the tick microbiome is different from species to species.

We are in agreement with reviewer that tick microbiome can be different from species to species. However, our preliminary comparative alpha diversity analysis and multivariate analyses of A. variegatum and A. hebraeum  microbiome revealed no significant differences (Figure A2). Therefore, for downstream analyses, A. variegatum and A. hebraeum samples were collectively grouped as Amblyomma ticks and compared with H. truncatum samples, herein referred to as Hyalomma ticks. This information has been clarified in sub section 2.1 (Line 143-148) and Appendix figure A2 (line 758-766). Where necessary, we have correctly referred to specific species or genera throughout the manuscript as suggested by the reviewer

Title: The title is too long. Please trim it and focus on the most relevant message.

We revised the manuscript title to be more specific as suggested by the reviewer  (Line 2-4)

Abstract:

Line 13: The expression ‘Deciphering ticks and their microbiomes’ needs revision. What does ‘Deciphering ticks and/or their microbiomes’ means?

Line 14: Change ‘tickborne’ to ‘tick-borne’ and revise this in the whole manuscript.

Lines 13-15: Sentence too long. Split it.

Line 16: ‘evolving drug resistance’ needs revision. It has no clear meaning.

Lines 18-19: Remove the full name of ‘PICRUSt’.

Line 20: ‘antibiotic resistance’ is trait intrinsic to some bacteria expressing ‘antibiotic resistance’ genes. In this study, the authors studied the presence of ‘antibiotic resistance’ functions in the microbiome of ticks. ‘antibiotic resistance’ lacks meaning here. Please, correct.

Lines 20-21: the fragment ‘antibiotic resistance in Amblyomma … and Hyalomma … microbiome in Nguni cattle’ needs revision. I suggest change to ‘antibiotic resistance in the microbiome of Amblyomma … and Hyalomma … ticks collected in Nguni cattle’.

Line 22: distinctions between ‘pathogenic and endosymbiotic bacteria’ cannot be made using the genera level as the taxonomic resolution. Please, revise. In almost any given bacterial genus (e.g., Rickettsia), you can find ‘pathogens, commensals and/or endosymbionts’. Almost nonbacterial genus has only ‘pathogens’.

Lines 21-25: In this sentence is not clear what bacterial taxa belongs to ‘Hyalomma’ and which one to ‘Amblyomma’. Also, as two ‘Amblyomma’ species were studied, please, change to ‘Amblyomma spp.  microbiomes’

Line 25: This study was not designed to identify ‘taxonomic biomarkers’. Remove

Line 29: What would the ‘key microbiome’ be? What the ‘non-key microbiome’ be?

The sentence has been corrected accordingly to be concise (line 12-13)

We have corrected this throughout the text as suggested

The sentence has been split to be more focused (line 19-22)

Sentence revised and evolving removed to give clear meaning. (line 15)

Corrected as suggested by the reviewer (line 15)

Sentence corrected to be precise as suggested by the reviewer (line 17)

Corrected as suggested by the reviewer (line 17-18)

We agree with the reviewer that same genus can have both pathogenic and non-pathogenic microbes. In this study , we could putatively identify both pathogenic and non-pathogenic species using EZbiocloud database (Figure A3). However, the only caveat is that the 16S rDNA fragment size of ~350-550bp is not sufficient to correctly identify bacteria to species level. In this manuscript, as a caution, we refer to microbes identified as both potential pathogens and/or endosymbionts (Line 18-19)

We have split the sentence and modified it to be concise (line 19-23)

The sentence has been corrected accordingly (line 22)

The sentence has been corrected accordingly (line 26)

Introduction

Line 39: Change ‘ticks of the genera Ixodes…’

Line 40: Doing what ‘in Nguni cattle’? infesting? Feeding on?

Line 41: Species the name of the pathogen species.

Line 42: The ‘R’ in Rickettsia goes in Italics too. Lines 47-49: Mention the species of the pathogens you are referring to. Line 51: Change ‘pathogens’ to ‘bacteria’. Lines 51-55:

This sentence is not the contraposition of the previous one, nor presents an exception from hat was stated in the previous one. Therefore, remove ‘However’. Instead, ‘On the other hand’ may be better.

Lines 57-59: Species should be named in full the first time they are mentioned in the text. Then, the abbreviated form can be used.

Lines 65-67: Change to ‘Despite increasing concerns of drug resistance development, there is limited information on the tick microbiome ecology and their involvement in drug resistance of tick-borne bacteria affecting local African cattle breeds.’

Lines 74-75: ‘barriers for … pathogen transmission’? Something that contributes to the transmission should not be regarded as a ‘barrier’.

Lines 91-101: It could be mentioned that previous studies used PICRUSt to study the tick microbiota should be mentioned in the introduction. What type of studies have used this methodology applied to tick microbiota? (e.g., see Pathogens. 2020 ;9(4):309. doi: 10.3390/pathogens9040309 and Microorganisms. 2020;8(11):1829. doi: 10.3390/microorganisms8111829.). Validation of PICRUSt predictions is also important. At least two studies have validated by PCR the functional predictions of PICRUSt on tick microbiome (Front Immunol. 2021;12:704621. doi: 10.3389/fimmu.2021.704621 and Vaccines (Basel). 2020;8(4):702. doi: 10.3390/vaccines8040702.). This further justifies the use of PICRUSt.

The sentence has been corrected as suggested (line 38)

We meant tick “infesting” Nguni cattle. The sentence has been corrected accordingly in line37

The sentence has been corrected accordingly in line38

The sentences have been corrected accordingly in line38-47

The sentences have been corrected accordingly in line49

The sentence has been corrected as suggested by the reviewer. line 55-57

Changed as advice line 63-65

The sentences have been corrected accordingly in line83

We have added more information on the studies on tick microbiome that have used both 16S rDNA and PICRUSt to deduce metagenomic functions as suggested by the reviewer (line 110-117)

Results

Line 165: ‘p’ in italics.

Line 167: Not clear why the authors performed this analysis at the ‘phylum’ level. Why not using a lower taxonomical category (e.g., genera)?

Lines 185-186: Percentage values can be organized from higher to lower or lower to higher. Also, taxonomic levels above ‘genera’ do not go in Italics (please revise this in the whole manuscript).

Lines 226-229: To what tick genera/species these results refer to.

Line 231: Here is important to specify the proportion of samples in which the ‘core’ taxa were found. It means that if, for example, only 1% of Amblyomma sp. samples have one given taxon, this cannot be considered as ‘core’ within that tick species. In other words, identification of a taxonomic core would require also presence across samples under study (see e.g., Pathogens. 2020 ;9(4):309. doi: 10.3390/pathogens9040309). In addition, defining a ‘core’ microbiome considering the small ‘n’ of the study is questionable.

Line 259: Here as in (line 22) please consider that distinctions between ‘pathogenic and endosymbiotic bacteria’ cannot be made using the genera level as the taxonomic resolution. Please, revise. In almost any given bacterial genus (e.g., Rickettsia), you can find ‘pathogens, commensals and/or endosymbionts’. Almost none bacterial genus has only ‘pathogens’.

Corrected accordingly (line 168)

The PCoA analysis was performed at genus level. We have corrected this accordingly in the legend of Figure 2 (line 181)

Corrected accordingly (line 186-188, 209-212)

The sentence has been modified to correctly illustrate which species were in the two tick genera/species (line 229-233)

We had defined what core taxa means “as OTUs present across at least 50% of the samples of each group and occurring at >1% relative abundance” as has been used by several studies (DOI: 10.1038/s41598-019-46388-1, https://doi.org/10.3389/fmicb.2021.775078, . https://doi.org/10.1371/journal. pone.0232398). However, our sample size may hamper authoritative identification of the core taxa. We have highlighted this as a limitation of the current study (line 590-599)

In this manuscript, as caution, we refer to microbes identified in the key genera as both potential pathogens and/or endosymbionts (Line 18-19, 263-277)

Discussion

The discussion (specifically from line 511) misses a crucial point of addressing the possibility that antibiotic resistance enzymes found in tick microbiome may be coded by genes that could be transferred to tick-borne bacteria via horizontal gene transfer or any other molecular mechanism accounting for genetic material transference in Prokaryotes.

We have improved the discussion to include the potential of antibiotic resistance gene transfer across the pathogenic taxa and endosymbionts of tick microbiome (line 575-587)

Material and methods

Figure 1. Change ‘Sample size’ to ‘Number of reads’.

Figure 3. The labels of the axis are not legible. Please, provide a corrected figure

Figure 4a. Despite the legend shows several taxa/colors, only four (blue, yellow, red and green) are visible in the bars. Then it is not clear the proportion of the four in taxa in relation to the others.

Figure 6. Genera names go in Italics.

Figure legends

Provide a short title for each figure and then follow with the explanation.

Changed the x-axis title to sequence counts in Figure A1 (line 750)

Corrected Figure provided Figure 2.

Corrected as Figure 3a for clarity

This has been corrected accordingly in Figure 5

This has been corrected accordingly in the Figures necessary (Figure 6, 7)

Reviewer 3 Report

I have read with interest paper authored by Chigwada et al, after that I have following suggestions and recommendations for the authors to consider during revisions:

L24 italic “and” is not needed

L 40, the title and in abstract please add Latin name of Nguni cattle, after first use of Latin name you can use only common name

Why the authors investigated the prevalence of Corynebacterium, Porphyromonas, Anaerococcus, Trueperella, Helcococcus? What is medical and/or veterinary importance of theses pathogens? In Intoduction section the authors describe Anaplasma, Coxiella, Borrelia, Rickettsia

Section 2.1 is more likely to methods than results. I suggest to move this part of ms and combine with section 4.

Figure 1 is not necessary

Figure 2 I suggest to change x axis label to Species. In what unit is alpha diversity measure shown?

Section 2.2. In my opinion it is not necessary to write what kind of statistical test was used in Results. This should be described and explained in Methods. Overall, in my opinion this section should be combined with Methods. The authors must explain what statistical test they used, and why? Now, I can see that they used “commonly used test” such as KS test, Wilcoxon rank-sum, but on the other hand they used statistical calculation performed in R. In current form of ms this question is not clear explained.

Figure 3. In my opinion the label of axis is wrong. What does mean “Axis.2 [13.1%]. The variable data should be labeled on axis.

Figure 4. what does mean abbreviations used by the authors, such as “OUT”?. What is shown in axis “x”? I suggest to label every column what bacteria species is shown. Part B of Figure 4 is unclear in its current form.

Figure 6 is not clear, thus I cannot judge its accuracy

Figures 8, 9 I sagest to move to supplementary file 

Author Response

Comments and Suggestions for Authors

I have read with interest paper authored by Chigwada et al, after that I have following suggestions and recommendations for the authors to consider during revisions:

Author Response: Thank you for recommending our work. We have supplemented the necessary contents and carefully revised the manuscript according to your suggestions, and we hope that the manuscript is now acceptable for publication in Pathogens.

L24 italic “and” is not needed

Corrected as suggested (line 17)

L 40, the title and in abstract please add Latin name of Nguni cattle, after first use of Latin name you can use only common name

Corrected as suggested (line 4 and Line 8).

We have also added additional information why Nguni cattle as an important cattle breed in Southern Africa in the introduction section (line 118-123)

Why the authors investigated the prevalence of Corynebacterium, Porphyromonas, Anaerococcus, Trueperella, Helcococcus? What is medical and/or veterinary importance of theses pathogens? In Intoduction section the authors describe Anaplasma, Coxiella, Borrelia, Rickettsia

Ticks not only carry pathogens but also a diverse group of commensal and symbiotic microorganisms. Unlike pathogens, the biology of non-pathogenic taxa and their effect on ticks remain largely unexplored, and are in fact often neglected. Nonetheless, they can confer multiple detrimental, neutral, or beneficial effects to their tick hosts, and can play various roles in the fitness, nutritional adaptation, development, reproduction, defense against environmental stress, and immunity to the tick host. Non-pathogenic microorganisms may also play a role in driving the transmission of tick-borne pathogens (TBP), with many potential implications for both human and animal health. (https://doi.org/10.3389/fcimb.2017.00236). Thus, we believe that deciphering the relationships between tick microorganisms as well as tick symbiont interactions will garner invaluable information than just considering the pathogens alone. Based on the above view, we have enriched the introduction section to highlight the importance of both pathogens and endosymbionts to tick microbiome fidelity and tick biology (line 68-78).

Section 2.1 is more likely to methods than results. I suggest to move this part of ms and combine with section 4.

We have modified the section merging it in materials and method section as suggested  (line 662-664, line 674-675)

Figure 1 is not necessary

The figure has been moved to Appendix A Figure A1 (line 739-752)

Figure 2 I suggest to change x axis label to Species. In what unit is alpha diversity measure shown?

The figure (now Figure 1) has been modified accordingly by removing the legend and x-axis titles which are redundant as the tick species name is already clearly defined in x-axis label. (line 157-164)

Section 2.2. In my opinion it is not necessary to write what kind of statistical test was used in Results. This should be described and explained in Methods. Overall, in my opinion this section should be combined with Methods. The authors must explain what statistical test they used, and why? Now, I can see that they used “commonly used test” such as KS test, Wilcoxon rank-sum, but on the other hand they used statistical calculation performed in R. In current form of ms this question is not clear explained.

Instances where the statistical test used were indicated in this subsection has been moved to materials and methods (section 2.4) which has been beefed up to clearly present the statistical test used for the study (line 676-695)

Figure 3. In my opinion the label of axis is wrong. What does mean “Axis.2 [13.1%]. The variable data should be labeled on axis.

The figure (now Figure 2) labels have been corrected and replotted as suggested  (line 177-183)

Figure 4. what does mean abbreviations used by the authors, such as “OUT”?. What is shown in axis “x”? I suggest to label every column what bacteria species is shown. Part B of Figure 4 is unclear in its current form.

The abbreviation is OTUs (Operational taxonomic unit) has been added on its first mention (line 137-138).

Figure 4a(now Figure 3a) displayed the relative abundance of the top phyla accounting for 99% of the sequence reads. We have corrected the legend to correctly depict the top phyla (line 197-198). In the x-axis A1-15 denotes Amblyomma tick samples while H1-3 denotes H. truncatum samples. This information has been added in Figure 4 legends (line 199-200)

The resolution of Figure 4b (now Figure 3b) (Line 197) has been improved for clarity

Figure 6 is not clear, thus I cannot judge its accuracy

The resolution of Figure 4b now Figure 5)(Line 254-262) has been improved for clarity

Figures 8, 9 I sagest to move to supplementary file 

One of the key objectives of this study was to infer the antibiotic resistance biomarkers in the microbiome associated with the tick species. The info contained in Figure 8 and 9 give a picture on the differential abundance of the key metabolic pathways and specific antibiotic resistance genes (ARG) ascribed to tick microbiomes. We believe this info is important as the metagenomic function assay provided clues for the main microbiome and their associated antibiotic resistance markers.

Round 2

Reviewer 2 Report

The authors addressed my previous concerns correctly.

Author Response

We are humbled and grateful for your efficient work the in-depth and insightful reviewing and processing of the draft that has greatly help to shape the article.

We hope the revised manuscript is now acceptable for publication in Pathogens 

Yours sincerely

Corresponding author: Aubrey Chigwada

Reviewer 3 Report

The authors have revised manuscript and I can agree with their responses. However, paper still requires minor revisions. The authors have to pay attention on sections numbers and figures labels, what figures are cited in the text of ms e.g. there is no section 2.2 and therefore other sections are wrong numbered. There is no figure label in line 286. Line 882 - there is section 2.4 - should be section 4.4 - the same flaw in Line 912. Also Fig.A2 is labeled twice ( Line 974 and 984).

The authors have to carefully check whole ms for similar flaws.

Author Response

Dear Editor/Reviewer,

We are still humbled and grateful for your efficient work and the reviewers in-depth and insightful suggestions on the draft that has greatly help to shape the article. We apologize for our previous carelessness of not correcting the figure citations and section numbering within the whole manuscript. We have corrected this and thoroughly checked for any typographical errors in the full text. All the changes made has been undertaken as tracked changes as suggested by the editor.

Figure citation and section numbering changes: line 230, 240, 271, 395, 442, 447, 508, 564, 568, 625, 626, 695,701,727, 736

Section numbering corrections: Line 288, 328, 339, 342,360,434, 884, 915

We hope the revised manuscript is now acceptable for publication in Pathogens and look forward to hearing from you at your earliest convenience.

Yours sincerely

Corresponding author: Aubrey Chigwada